# A Multi-Omics Approach Identifies Key Regulatory Pathways Induced by Long-Term Zinc Supplementation in Human Primary Retinal Pigment Epithelium

**DOI:** 10.3390/nu12103051

**Published:** 2020-10-06

**Authors:** Eszter Emri, Elod Kortvely, Sascha Dammeier, Franziska Klose, David Simpson, Anneke I. den Hollander, Marius Ueffing, Imre Lengyel

**Affiliations:** 1Wellcome Wolfson Institute for Experimental Medicine, Queen’s University of Belfast, Belfast BT97BL, Northern Ireland, UK; e.emri@qub.ac.uk (E.E.); david.simpson@qub.ac.uk (D.S.); 2Roche Pharma Research and Early Development, Immunology, Infectious Diseases and Ophthalmology (I2O) Discovery and Translational Area, Roche Innovation Center Basel, F. Hoffmann-La Roche Ltd., 4070 Basel, Switzerland; elod.koertvely@roche.com; 3Institute for Ophthalmic Research, University of Tubingen, D-72076 Tubingen, Germany; sascha.dammeier@web.de (S.D.); franziska.klose@klinikum.uni-tuebingen.de (F.K.); marius.ueffing@uni-tuebingen.de (M.U.); 4Departments of Ophthalmology and Genetics, Radboud University Medical Center, 6525EX Nijmegen, The Netherlands; anneke.denhollander@radboudumc.nl

**Keywords:** zinc, retinal pigment epithelium, age-related macular degeneration, transcriptome, proteome, secretome, gene set enrichment, TGFB1

## Abstract

In age-related macular degeneration (AMD), both systemic and local zinc levels decline. Elevation of zinc in clinical studies delayed the progression to end-stage AMD. However, the molecular pathways underpinning this beneficial effect are not yet identified. In this study, we used differentiated primary human fetal retinal pigment epithelium (RPE) cultures and long-term zinc supplementation to carry out a combined transcriptome, proteome and secretome analysis from three genetically different human donors. After combining significant differences, we identified the complex molecular networks using Database for Annotation, Visualization and Integrated Discovery (DAVID) and Ingenuity Pathway Analysis (IPA). The cell cultures from the three donors showed extensive pigmentation, development of microvilli and basal infoldings and responded to zinc supplementation with an increase in transepithelial electrical resistance (TEER) (apical supplementation: 443.2 ± 79.3%, basal supplementation: 424.9 ± 116.8%, compared to control: 317.5 ± 98.2%). Significant changes were observed in the expression of 1044 genes, 151 cellular proteins and 124 secreted proteins. Gene set enrichment analysis revealed changes in specific molecular pathways related to cell adhesion/polarity, extracellular matrix organization, protein processing/transport, and oxidative stress response by zinc and identified a key upstream regulator effect similar to that of TGFB1.

## 1. Introduction

Age-related macular degeneration (AMD) is a progressive, multi-factorial disease that leads to irreversible vision loss [1,2,3,4]. It has no cure, and only limited treatment options are available to delay, rather than prevent, the progression to end-stage AMD. Dietary intake of zinc has shown promise in delaying the progression of AMD [5,6,7].

Zinc is an essential micronutrient, critical for normal cellular processes in all organisms [8]. It acts as a catalyst for more than 300 enzymes, and thousands of proteins require zinc for their normal structure and function [9,10,11,12,13,14,15,16,17,18,19,20,21]. The eye has an unusually high content of zinc, primarily in the retinal pigment epithelium (RPE) [21,22]. The RPE forms a monolayer of highly specialized pigmented cells located between the neurosensory retina and the choroidal vasculature and plays a pivotal role in the visual cycle and maintaining the health of the outer retina [23]. Consequently, extensive research efforts have focused on the cell biology of the RPE [24,25,26,27,28,29,30,31,32,33,34,35] including studies examining the effects of zinc supplementation [36,37,38]. These studies resulted in mixed outcomes [12,39,40,41,42,43] likely due to the availability of zinc to exert biological function being affected by differing compositions of culture media [44]. Recently, we established the optimal conditions to study the effects of zinc on human RPE cells [38] and in this paper, we set out to identify the key regulatory pathways involved in long-term zinc supplementation by combining the zinc-induced changes in cell transcriptome and proteome and apical and basal secretomes. In this study, we identified specific regulatory pathways that might be involved in mediating the positive effects of long-term zinc supplementation in AMD.

## 2. Materials and Methods

### 2.1. Retinal Pigment Epithelial (RPE) Cell Culture

Primary human fetal RPE cells (ScienCell, Carlsbad, CA, USA) from three donors with unknown clinical or genetic background were used for our experiments at passage number three (P3). Cells were seeded onto Corning 6-well transwell inserts (10 µm thick polyester inserts with 0.4 µm pore size, 4*10^6^/cm^2^ pore density, Corning, Wiesbaden, Germany) in epithelial cell medium (EpiCM, ScienCell, Carlsbad, CA, USA). After seven days, the cell culture medium was replaced with Miller medium with 1% FBS [45,46] for an additional four weeks in the absence or presence of 125 µM externally added zinc (as zinc sulphate; Thermo Fisher Scientific, Waltham, MA, USA), resulting in ~10 nM bio-available or free zinc [38,44]. Cellular differentiation was monitored through the development of cobblestone morphology and increase in pigmentation using light microscopy and the increase in transepithelial electrical resistance (TEER) was measured by using the EVOM2 Epithelial Voltohmmeter and STX2 electrodes (World Precision Instruments, Sarasota, FL, USA). The obtained results were analyzed using two-way ANOVA and Tukey’s test and a *p*-value < 0.05 was considered significant.

At the end of the experiments, cells were washed with PBS (Thermo Fisher Scientific, Waltham, MA, USA) 3 times for 2 min. Then differentiation media were replaced with serum-free Minimum Essential Medium Eagle Alpha Modifications (Merck, Darmstadt, Germany). After 24 h, apical and basal media (approximately 600 and 900 µl, respectively) were collected, snap frozen in liquid nitrogen and stored at −80 °C until proteomics analysis was conducted. The transwell membranes containing the cells were excised and divided into five equal-sized wedges. These were either immediately snap frozen and kept at −80 °C or fixed for 15 min in 4% PFE (Merck, Darmstadt, Germany) diluted in PBS (Thermo Fisher Scientific, Waltham, MA, USA) or kept in Karnovsky fixative (Agar, Essex, UK) comprising 3% (*v/v*) glutaraldehyde, 1% (*v/v*) paraformaldehyde in 0.08 M sodium cacodylate buffered to pH 7.4 with 0.1M HCl.

#### 2.1.1. Immunofluorescence

For immunofluorescence analysis, the cells on the transwell membrane were permeabilized in 0.5% Triton-X (Merck, Darmstadt, Germany) in PBS for 10 min at 4 °C and then washed in 0.1% Tween20 in PBS (PBST) (Merck, Darmstadt, Germany) and blocked with PBST with 5% goat sera (Merck, Darmstadt, Germany) for one hour at room temperature. Next, samples were incubated with primary antibodies PMEL17 (Agilent, Santa Clara, CA, USA, dilution 1:25), ZO-1 (BD Biosciences, San Jose, CA, USA, 1:200), BEST-1 (Merck, Darmstadt, Germany, 1:50), and RPE65 (Merck Millipore, Darmstadt, Germany, 1:50) diluted in PBST containing 1% goat sera. Following washing with PBST, the samples were incubated with secondary antibodies in 1:200 in PBST with 1% goat sera for one hour in the dark at room temperature. After, the secondary antibody incubation samples were washed with PBST for 5 min, followed by 5 min washing with PBS. Cell nuclei were then labelled for 15 min with DAPI (Thermo Fisher Scientific, Waltham, MA, USA) diluted 1:1,000 in PBS. Finally, the samples were washed for 5 min in PBS, before mounting onto Menzel-Glaser slides (Thermo Fisher Scientific, Waltham, MA, USA) in Vectashield (Vector Laboratories, Burlingame, CA, USA). The flat mounts were sealed by applying nail polish around coverslips. For negative control, the primary antibody labelling was replaced by incubation with PBST only. Results were visualized using a Leica SP8 confocal microscope (Leica, Wetzlar, Germany) with a 40x/1.25 oil immersion objective. Images were obtained and analyzed with Leica Application Suite X Image software (Leica, Wetzlar, Germany).

#### 2.1.2. Transmission Electron Microscopy

For transmission electron microscopy, the 1% glutaraldehyde (Agar, Essex, UK) and 2.5% paraformaldehyde-fixed samples were post-fixed in 1% (*w*/*v*) osmium tetroxide (Agar, Essex, UK) in 0.1 M PBS for 50 min, dehydrated and embedded in Araldite (Agar, Essex, UK). Semi-thin sections were generated using Leica ultra-microtome (Leica, Wetzlar, Germany), stained with 1% (*w*/*v*) uranyl acetate and Reynolds’ lead citrate and viewed with a JEOL JEM-1010 electron microscope (JEOL USA, Peabody, MA, USA) and a Gatan Orius CCD camera (Gatan, Pleasanton, CA, USA).

### 2.2. Genotyping

For genotyping, genomic DNA was extracted from snap-frozen transwell inserts using AllPrep DNA/RNA/Protein Mini Kit (Qiagen) following manufacturers’ instructions. The EYE-RISK genotyping assay [47] was used to genotype 87 single-nucleotide polymorphisms (SNPs), including the 52 independently AMD-associated SNPs identified by the International AMD Genomics Consortium [2], SNPs previously associated with AMD [48], and several candidate SNPs. For nine of the 52 AMD-associated SNPs, an alternative SNP in high linkage disequilibrium (≥0.80) was used. Furthermore, the coding and splice-site regions of thirteen genes (*C3, C9, CFH, CFI, TIMP3, SLC16A8, ARMS2, CD46, CFB, HTRA1, ABCA4, CTNNA1,* and *PRPH2*) were completely sequenced. The targeted regions were enriched by single-molecule molecular inversion probes (smMIPs) followed by next-generation sequencing of the library pools, as described previously [47]. 

A genetic risk score (GRS) for the 52 AMD-associated SNPs [2] was calculated based on the formula $GRS=\;\sum_{i=1}^{52}\left(G_{i\;}\beta_{i}\right)$, as described previously [47]: $G_{1}$ represents the genotype of variant i, where genotypes were coded as 0, 1 or 2 based on the number of minor alleles (0 = carrier of 0 minor alleles, 1 = carrier of one minor allele, 2 = carrier of two minor alleles). $\beta_{1}$ represents the effect size of variant i (natural logarithm of the fully conditioned odds ratio [OR] of the minor allele of variant i), based on the genome-wide association study (GWAS) of the International Age-related Macular Degeneration Genomics Consortium (IAMDGC) [2].

### 2.3. Multi-Omics Approach Development

#### 2.3.1. Transcriptomics

For transcriptomic analysis, total RNA was extracted simultaneously with genomic DNA isolation from snap-frozen transwell inserts using a AllPrep DNA/RNA/Protein Mini Kit (Qiagen) following manufacturers’ instructions. The quality of RNA was analyzed using the AATI Fragment Analyzer and Qubit (Thermo Fisher). Libraries were prepared by the QUB Genomics Core technology Unit using the KAPA HyperPrep kit. Libraries were quantified, normalized and pooled in equimolar amounts and sequenced on the Illumina NextSeq 5000. QC was performed with Qualimap following alignment to human genome build GRCh37 (hg19). Raw counts calculated using htseq were analyzed further for differentially expressed genes (DEG) performing quasi-likelihood F-test in edgeR package [49]. To filter out lowly expressed genes, the following script was used: > keep <- rowSums(cpm(y)> 0.05) > = 1. To be considered as DEG, the Benjamini–Hochberg-adjusted P-value was less than 0.05. The full list and raw data are deposited in the GEO database with dataset identifier GSE156452.

#### 2.3.2. Proteomic Sample Preparation

For proteomic analysis, all solvents and water were purchased from Merck (Darmstadt, Germany) as HPLC-grade purity. All chemicals were purchased from Sigma Aldrich (Taufkirchen, Germany) unless stated otherwise. HPLC columns were obtained from Thermo Fisher Scientific (Dreieich, Germany).

To analyze the secretomes of the RPE transwell cell culture, apical and basal supernatants were thawed on ice and concentrated to approximately 100 µL volume by evaporation. To generate sample fractions for optional metabolomic and lipidomic analyses, in addition to proteomics the SIMPLEX extraction method was applied according to the protocol described elsewhere [50]. Finally, the methanol-containing phase was removed, and the protein pellet was air dried and frozen at −80 °C until further analysis.

For cellular proteomic analysis, 200 µL of cold lysis buffer (Tris-buffered saline containing 1% NP-40 and 2% cOmplete^TM^ protease inhibitor cocktail (Merck, Darmstadt, Germany/Roche, Basel, Switzerland)) were added to one snap-frozen wedge of each transwell chamber, and the cells were detached on ice using a cell pestle (Merck, Darmstadt, Germany, # Z359947-100EA). The resulting suspension was vortexed and subsequently centrifuged at 16,000 × g at 4 °C for 10 min. The clear supernatant was recovered, and the protein concentration was determined using a conventional assay (Bradford, Biorad, California, USA). For each sample, a volume containing 10 mg of total protein was used to perform a methanol/chloroform precipitation. The resulting protein pellet was air dried and used for proteome analysis as described in the following section.

#### 2.3.3. Proteome Analysis by Mass Spectrometry

Frozen protein pellets were recovered in 30 µL of 50 mM ammonium bicarbonate solution plus an additional 4 µL of RapiGest SF Surfactant (Waters, Eschborn, Germany). Proteins were reduced by adding dithiothreitol followed by an incubation with iodacetamide to alkylate cysteine residues. Finally, limited proteolysis was performed by adding 1 µL of 0.5 µg/µl trypsin (Merck, Darmstadt, Germany) and incubation at 37 °C for 12 h. The reaction was stopped by adding trifluoroacetic acid to a final concentration of 5%. After centrifugation, the supernatants were processed using StageTips (Thermo Fisher Scientific, Waltham, MA, USA) according to the manufacturer’s protocol. LC–MS/MS analysis was performed on a NanoRSLC3000 HPLC system (Dionex, Thermo Fisher Scientific, Waltham, MA, USA) coupled to an Orbitrap Fusion mass spectrometer (Thermo Fisher Scientific, Waltham, MA, USA) by a nano spray ion source. Tryptic peptide mixtures were trapped on a nano trap column (75 μm i.d. × 2 cm, packed with Acclaim PepMap100 C18, 3 μm, 100 Å; Dionex, Thermo Fisher Scientific, Waltham, MA, USA), and separated on an analytical column (75 μm i.d. × 25 cm, Acclaim PepMap RSLC C18, 2 μm, 100 Å; Dionex, Thermo Fisher Scientific, Waltham, MA, USA) applying a gradient of actetonitrile in water with 0.1% formic acid.

For quantitative analysis, MS raw data were processed using the MaxQuant software (version 1.5.3.30) [51]. Two missed cleavages per peptide were allowed. The peptide and protein false discovery rates were set to 1%. The initial mass tolerance for precursor ions was set to 4.5 ppm and the first search option was enabled with 20 ppm precursor mass tolerance. The fragment ion mass tolerance was set to 0.5 Da. The SwissProt_2014 database (selected for homo sapiens, 20203 entries) was used for peptide and protein identification. The MaxQuant software-generated output file “proteingroups.txt” was utilized for Pearson correlation, clustering and statistical analysis using Perseus software version 1.5.5.3. All data were filtered for potential contaminants, peptides only identified by side or reverse sequence. The dataset was filtered further on Log2 transformed LFQ values to the minimum number of valid values “in at least one in total”, and missing values were imputed using normal distribution in “separately for each column” mode. All numeric values were expressed as the mean ± standard deviation (SD) for the respective groups. Benjamini–Hochberg corrections of significance B calculated by the Perseus software package were used to identify the intensity-dependent significant outliers relative to a certain populations [51]. The adjusted p-value of less than 0.05 was considered a significantly enriched protein. The mass spectrometry proteomics data have been deposited to the ProteomeXchange Consortium via the PRIDE [52] partner repository with the dataset identifier PXD020860.

#### 2.3.4. Functional Classification Pathway and Network Analysis

Gene ontology (GO) analysis, canonical pathway and network identification were performed using Database for Annotation, Visualization and Integrated Discovery (DAVID) Bioinformatics Resources 6.8 with the Kyoto Encyclopedia of Genes and Genomes (KEGG) database [53,54], with p-values of less than 0.05 considered as significant enrichment. In addition, we used the Ingenuity Pathway Analysis (IPA 7.4, April 2009) with the HumanCyc database, with the default threshold of IPA, a cut-off *p*-value of -log (*p*-value) < 1.3 used for statistically significant enrichment for top canonical pathways. For overlapping canonical pathway visualization, the network was filtered for 250 significant pathways (*p*-value < 0.05) with a minimum of three common shared molecules. A total of 67 entries exported from relevant GWAS studies [2,55,56] were used in IPA for overlaying the dataset on AMD-relevant molecules.

## 3. Results

### 3.1. Characterization of Human Primary RPE Cell Culture

Based on previous experiments [34,38], we used conditions under which RPE cells in culture developed dense microvilli (Appendix A), basal infoldings (Appendix A), extensive pigmentation that is localized to the apical aspects of the cells (Appendix A) with pigment particles appearing in the microvilli (Appendix A), basal lamina (Appendix A; red arrowheads) and expressed RPE-specific proteins such as PMEL17, ZO-1, BEST-1 and RPE65 (Appendix A, respectively). In addition, sub-RPE deposits are visible (Appendix A) similar to those of mature RPE in vivo [57,58].

While the cells from the three individuals all showed similar phenotypes (see Appendix A), there were differences between the cultured cells. Specifically, there were differences in their pigmentation pattern (Figure 1A (donor (D)1-3)), the numbers of visible sub-RPE deposits (Figure 1A (D1-3, labelled by red arrowheads) and the maximal TEER values reached after 5 weeks in culture (Figure 1B (D1: 477.9 ± 28.4, D2: 244.4 ± 45 and D3: 1114.6 ± 27 Ohm × cm^2^)).

The genotypes of the 52 AMD-associated SNPs and rare variants in 13 genes identified in the three donors are provided in Appendix A (genotypes different between donors are highlighted in bold). The donors had a relatively low genetic risk score (GRS) based on the 52 AMD-associated SNPs: donor 1 has a GRS of −0.33; donor 2 has a GRS of −2.36; and donor 3 has a GRS of 0.95 compared to the GRS of AMD patients [47]. The donors carried different genotypes for the major *CFH* SNP rs10922109 (donor 1 CA, donor 2 AA, donor 3 CC), and carried the same low-risk genotype for the major *ARMS2* SNP rs10490924 (all three donors GG). In addition, donor 3 carries a rare, heterozygous variant in the *CFI* gene (c.299C > T; p.Phe100Ser) of unknown clinical significance (Appendix A, genotypes different between donors are highlighted in bold).

### 3.2. Zinc and RPE Barrier Function

Despite their differences, the cell cultures responded to zinc supplementation similarly with respect to changes in transepithelial electrical resistance (TEER) values (Figure 1C), irrespective of whether zinc was added apically or basally. Due to the differences in the maximum TEER values (Figure 1B), the zinc-induced changes in barrier function were normalized to TEER values measured at day 4 (Figure 1C). After this normalization, we found that either apical or basal zinc led to a significant increase in the TEER values from day 16 and this increase remained significant for the remainder of the experiment (at week 4; control: increase of 317.5 ± 98.2%, zinc basal: increase of 424.9 ± 116.8%, zinc apical: increase of 443.2 ± 79.3% relative to day 4).

### 3.3. Multi-Omics Approach for the Effects of Zinc on RPE

After establishing that the cell cultures from the three donors share many phenotypic features characteristic of differentiated RPE and responded similarly to zinc treatment in terms of TEER measurement, we harvested samples for our multi-omics analysis as depicted in Figure 1D at the end of week 5 in culture.

#### 3.3.1. Cellular Transcriptome

To perform transcriptomic analysis, RNA was isolated at the end of the week 5 culturing period from the triplicate experiments. Using RNAseq, we were able to identify over 32,000 transcripts. A total of 826 of these were significantly changed in the three independent samples after apical and 216 after basal zinc supplementation when compared to untreated controls (Figure 2A, B, adjusted *p*-value < 0.05). Apical zinc supplementation upregulated 415 and downregulated 411 transcripts (Figure 2A red and blue dots, respectively). Basal zinc supplementation upregulated 157 and downregulated 59 transcripts (Figure 2B). A total of 163 transcripts changed similarly after both apical and basal treatment (for the full list of significantly changed transcripts see Appendix A).

#### 3.3.2. Cellular Proteome

In parallel to RNA isolation, we also isolated the protein fraction from the cells. We were able to identify 2164 proteins in the RPE cells. Following apical zinc supplementation, significant changes were observed in 89 (adjusted *p*-value < 0.05, Figure 2C, Appendix A). Basal application of zinc resulted in significant changes in 62 proteins (adjusted *p*-value < 0.05, Figure 2D, red dots, Appendix A). A total of 45 proteins changed similarly after both treatments.

There was an overlap in the mRNAs and proteins altered following apical supplementation of zinc (CSRP1: 1.29 and 2.52, CRIP2: 0.76 and 1.78, TTR: -0.65 and 2.6 log_2_ fold change at the transcriptomic and proteomic levels, respectively, Appendix A). After basal zinc supplementation, only CSRP1 was significantly changed in both transcriptome and proteome (CSRP1: 1.1 and 2.52 log_2_ fold change, at the transcriptomic and proteomic levels, respectively, Appendix A).

#### 3.3.3. Apical and Basal Secretomes

As many proteins changed following zinc supplementation could have been secreted from cells, we collected apical and basal media for the analysis of protein secretion. We identified 1132 proteins in the apical secretomes, of which 33 changed significantly to apical and 37 to basal zinc supplementation (Figure 2E,F respectively, adjusted *p*-value < 0.05). In the basal secretome, we identified 839 proteins from which 34 responded to apical (Figure 2G) and 20 to basal supplementation (Figure 2H, adjusted *p*-value < 0.05). Twenty-one proteins changed similarly in the apical secretome and 9 in their basal secretome after the treatments. The lists of proteins are provided in Appendix A.

Examining the overlaps between the changes in cellular transcriptome and apical secretome after apical zinc supplementation, we found two overlaps: STC2 (0.35 and 3.53 log_2_ fold change at the transcriptomic and apical secretome levels, respectively) and TPM1 (0.47 and −2.04 log2 fold change at the transcriptomic and apical secretome levels, respectively). There were two overlaps between cellular proteome and apical secretome: PSMA3 (2.04 and −3.05 log2 fold change at the proteomic and apical secretome levels, respectively) and COL12A1 (1.59 and −2.61 log2 fold change at the proteomic and apical secretome levels, respectively) (Appendix A).

Examining the overlaps between the cellular transcriptome and apical secretome after basal zinc supplementation, we found one overlap: STC2 (5.56 and 3.84 log2 fold change at the transcriptomic and apical secretome levels, respectively). There were two overlaps between cellular proteome and apical secretome: RTN4 (1.51 and −4.2 log2 fold change at the proteomic and apical secretome levels, respectively) and SERPINB12 (2.58 and 2.3 log2 fold change at the proteomic and apical secretome levels, respectively) (Appendix A).

Examining the overlaps between cellular transcriptome and, this time, basal secretome after apical zinc supplementation, we found two overlaps: MYL6 (0.37 and 2.65 log2 fold change at the transcriptomic and basal secretome levels, respectively) and SELENBP1 (−0.47 and −3.7 log2 fold change at the transcriptomic and basal secretome levels, respectively). There were three overlaps between the cellular proteome and the basal secretome: ARG1 (2.36 and 3.18 log2 fold change at the proteomic and basal secretome levels, respectively), DBI (−2.02 and 2.9 log2 fold change at the proteomic and basal secretome levels, respectively) and SERPINB12 (3.6 and −4.34 log2 fold change at the proteomic and basal secretome levels, respectively) (Appendix A).

When zinc was supplemented basally, there were no overlaps between the significantly changed cellular transcriptome and secreted proteins. When the cellular proteome and the basal secretomes were compared, the only significant change was observed in ARG1 (3.05 and 3.8 log2 fold change for, respectively, Appendix A).

### 3.4. Pathway Identification

To evaluate the complex impact of zinc supplementation on RPE cells, we combined the significant changes in transcriptome, intracellular proteome and the basal and apical secretomes following either basal or apical zinc supplementation and analyzed the data by DAVID and IPA. We converted the protein symbols to gene symbols for the analysis, which resulted in 983 entries for apical and 335 entries for basal zinc treatment.

#### 3.4.1. DAVID Gene Ontology (GO) and KEGG Pathway Analysis

The gene lists were first input into DAVID to identify biological processes, the involvement of cellular components, molecular functions and KEGG pathways that were significantly affected by zinc supplementation (the full list of the potential biological impacts is in Appendix A). Table 1 shows the top five most significant hits following zinc supplementation (*p*-value < 0.05).

#### 3.4.2. IPA Canonical Pathways and Regulatory Effect Networks

Following the analysis in DAVID, we set out to identify the canonical pathways, upstream regulators, disease associations and networks underpinning the effects of zinc supplementation by entering the data in IPA that potentially increases the statistical power of our pathway analysis by using an additional background database (summary of the analysis is Appendix A). Table 2 shows the top five hits identified by IPA following zinc supplementation (*p*-value < 0.05).

By focusing on the top canonical pathways, we were able to identify 70 pathways after apical and 48 pathways after basal zinc supplementation (−log(*p*-value) < 1.3, Appendix A). There were 28 common pathways after apical and basal zinc supplementation (Appendix A).

Ostensibly unrelated canonical pathways can be interconnected through shared molecules represented by gene names. Here we determined the network of overlapping canonical pathways that shared at least three genes for both apical (Figure 3A) and basal (Figure 3B) supplementation (Appendix A). In Figure 3, each canonical pathway is shown as a single “node” (the brighter red, the more significant the pathway).

Overlapping pathways identified five distinct clusters of which there were four small clusters (pyrimidine ribonucleotides de novo biosynthesis and interconversion; citrulline biosynthesis and metabolism; IL-23 and Th17 activation; phosphoinositide biosynthesis, metabolism and degradation) and one large cluster (the full list of canonical pathways are listed in Appendix A) with several highly interconnected canonical pathways after apical zinc supplementation (Figure 3A).

Following basal zinc supplementation, we identified three separate clusters with two small clusters (GP6 and hepatic fibrosis/hepatic stellate cell activation; super pathway of inositol phosphate compounds; and phosphoinositide degradation) and one large cluster (the full list of canonical pathways are listed in Appendix A) with multiple interconnected canonical pathways (Figure 3B).

Within the large clusters, we identified eight groups after apical supplementation and five groups after basal supplementation (groups are demarcated by black squares in Figure 3; Appendix A). The five groups identified after basal supplementation are all present after apical supplementation (adhesion to polarity, protein export/import, extracellular matrix organization, protein processing and oxidative stress). Three unique groups (Groups #6, #7 and #8, embryonic cell development, lipid metabolism and immune response) were identified after apical supplementation.

Regulator effects explain how predicted activated or inhibited upstream regulators might cause increases or decreases in phenotypic or functional outcomes downstream (for more details, see [59]). The regulator effects enrichment analysis in IPA predicted one directionally coherent network with TGFB1 as potential upstream regulator effect, but only when cells were supplemented with apical zinc (Figure 4). VIM, MMP2, CDKN1A, SERPINA1, F3 (also known as Tissue Factor), miR-145, and FAS dataset genes were identified as TGFB1-targeted downstream master regulators. The predicted disease mechanism was the inhibition of progressive neurological disorder (Figure 4). There was no regulator effect enrichment identified after basal zinc application.

## 4. Discussion

Identification of molecular pathways involved in the progression to late stages of AMD is vital to develop new or improved intervention strategies to stop the development of irreversible visual loss. One of the proposed intervention strategies for AMD is to modify the intake of a group of molecules that includes zinc [60]. Based on the Rotterdam study [61], Blue Mountains Eye study [62] and AREDS study [60], restoring zinc balance can slow the progression to end-stage AMD [5,6,7,63]. However, the molecular networks involved in this beneficial effect are not fully understood. In this work, we identified potential key regulatory pathways through which zinc may exert its positive effects in vivo.

To study the effects of zinc supplementation, we used a human fetal primary RPE cell culture model for AMD. This model has been shown to recapitulate aspects of early AMD including deposition of sub-RPE material [34]. We followed protocols from these earlier studies to ensure best culturing conditions and that sufficient zinc is available for the culture without causing cell toxicity [38,44].

While chronic exposure to elevated zinc reproduced the increased TEER and enhanced cell pigmentation as reported earlier [38], there were significant individual differences between maximal TEER values, levels of pigmentation and numbers of sub-RPE deposits in the primary RPE cell cultures derived from three different donors (Figure 1). These differences are likely to reflect genetic and phenotypic variations between individual donors from whom the cells were isolated from and may reflect the variability among individuals in real life. These individual differences also suggest that using precision medicine in the treatment of AMD can result in immense patient benefits.

Chronic treatment usually results in homeostatic cellular re-balancing [64] which may explain the observed moderate fold changes in our experiments. However, the long-term supplementation is aimed to reflect the long-term influence of zinc nutrition or supplementation used in population based studies [5,6,7,63].

In our previous study, we only applied zinc apically [38]. However, in vivo zinc supplementation would likely affect both basal and apical zinc availability. In this study, the basal elevation of zinc was designed to mimic changes associated with altered zinc concentration in the sub-RPE space, probably through transport from the systemic circulation via the choroidal vessels, while the apical supplementation was designed to replicate changes in zinc concentration in the sub-retinal space, where the photoreceptors and the interphotoreceptor matrix form an interdigitated layer with the apical surface of the RPE.

To evaluate the complex effects of zinc supplementation, we combined the significant changes observed in the transcriptome, proteome and secretomes of RPE after either apical or basal supplementation and performed analysis using DAVID and IPA for pathway identification. Both analyses revealed five main biological functions shared between apical and basal zinc supplementation: oxidative stress response, protein processing, protein export/import, adhesion to polarity and extracellular matrix organization (summarized with grouping of overlapping canonical pathways in Appendix A). Zinc is known to affect oxidative processes by modulating the expression of metallothioneins [65,66], glutathione [67,68,69], and catalase [12,42], by stabilizing sulfhydryl groups on proteins [70] and by replacing redox-active metals, such as copper and iron [71,72]. With ageing in general and in AMD in particular, protein, lipid and mineral-rich deposits accumulate in the sub-RPE space. The effects on protein processing and protein export/import in this study suggest that zinc supplementation might modulate directly sub-RPE accumulation. The effects of zinc in sub-RPE deposit formation in clinical studies are varied (see review [73,74]) and will require further investigations, probably with the use of methods that are able to specifically identify sub-RPE deposits in a clinical setting [75,76,77]. Changes in cell adhesion to polarity [78] and modulation of extracellular matrix organization had also been reported in AMD [31,79] which highlight another layer of complexity of zinc action on the RPE. As RPE activation and migration are significant clinical signs of the progression of AMD [80], this finding could point to new studies in which the clinical effects of zinc could be evaluated based on these new clinical endpoints. Overall, these findings highlight that zinc supplementation or increased nutrient intake have diverse effects on the RPE. Given that both apical and basal supplementation triggered these biological effects, increasing zinc concentration in the retina or in the blood circulation appears to be beneficial in AMD.

However, there were several biological functions affected specifically following apical zinc supplementation in our experiments. It is not yet known how the availability of zinc is regulated in health or in disease in the subretinal space. In the photoreceptor outer segments there is a high demand for zinc for the normal function of opsins [81,82,83]. This demand could be served by the release of zinc from the zinc-rich RPE [84], transport through the RPE from the choroid via zinc specific transport mechanisms [85,86] or release of zinc from the Müller glia [87]. As availability of apical zinc is critical for maintaining the high demand for the visual cycle [81,82,83,88,89,90,91] examining effects of increased apical zinc further.

The biological functions affected specifically by apical zinc supplementation were embryonic cell development, immune response and lipid metabolism. Differentiation and dedifferentiation of RPE had been linked with AMD [92]. Zinc is a known effector of infection and immunity via NF-kappaB activation and its targets such as TNF-alpha, IL-1beta, A20 and PPAR-alpha and inhibition of cyclic nucleotide phosphodiesterase [71,93] and as inflammation is strongly associated with AMD [57,94,95,96] our findings suggest that further clinical investigations are needed to uncover the relationship between zinc and the immune function. In addition, disturbances in lipid metabolism, both genetically and functionally, in AMD are well documented (for review see [97]). Therefore, these specific apical effects observed in our study suggest potential targets of zinc supplementation that could be studied further to identify new mediators for the progression of AMD in vivo.

We also investigated how the effect of zinc might directly link with the genes whose polymorphisms have been independently associated with AMD [2,55,56]. From these 67 AMD-associated genes, apical zinc supplementation influenced 6, and basal zinc supplementation affected one gene (Appendix A). Amongst these, we found a significantly increased expression of CFI following basal zinc supplementation. Zinc supplementation in AMD patients has been shown to reduce the activity of the complement system [98] and our results on CFI suggest that this reduced activity might be due to the increased expression of CFI. Another regulator of the complement system is CFH, one of the most significantly associated gene in relationship to AMD [2,55]. In our study, we found inconsistent zinc induced changes in both *CFH* transcripts, intracellular or secreted protein levels which might be due to low sample size and our donor to donor variability in the *CFH* genotype (Appendix A). Another major AMD gene association is related to HTRA1/ARMS2, and we found a significant increase in cellular HTRA1 protein levels, although there was no significant change observed in the transcriptome or apical or basal secretomes (Appendix A). Whether the significant cellular change is related to the fact that all three donors were homozygous for the low-risk *ARMS2* rs10490924 and *HTRA1* rs11200638 genotype needs further examination on larger samples sizes. The direct clinical implications of zinc supplementation regarding these genes are highlighted in the recent controversy regarding the role of zinc in AMD progression in the AREDS study. Awh et al. [99] reported that patients with two *CFH* risk alleles and no *ARMS2* alleles progressed more frequently to late AMD when supplemented with zinc, while those with no or just one *CFH* risk allele and one or two *ARMS2* risk alleles did benefit from zinc supplementation. These results, however, could not be replicated by Chew et al. [100]. Together with our data, further examination of zinc and genotype/protein activity might be highly relevant for AMD patients.

Regulator effects analysis in IPA enables generation of a hypothesis for how a phenotype, function or disease is regulated in a dataset by activated or inhibited upstream regulators. While there were no regulator effects predicted for basal zinc supplementation, apical zinc supplementation identified TGFB as a predicted master regulator (Figure 4). There is an emerging role for TGFB signaling in AMD [101,102] partly due to its role in angiogenesis [103]. Interestingly, its role in AMD is likely to be related to its receptor (TGFBR1) in RPE rather than the TGFB itself [101,102]. TGFB secreted by choroidal macrophages [102] seems to elicit secretion of VEGF by RPE leading to changes in vascular behavior [102]. While TGFB was expressed in our model, confirming a previous report [104], its levels were not modulated by changing zinc. However, downstream targets for TGFB, such as VIM (vimentin), MMP2, CDKN1A, SERPINA1, miR-145, F3 (tissue factor) and FAS, were all modulated by apical zinc supplementation (Figure 4). Vimentin directly interact with zinc and this interaction affects its assembly as well as redox sensing [105]. Due to its association with epithelial-mesenchymal transition, pigmentary abnormalities as well as the migration of RPE, vimentin also plays a role in the pathogenesis of AMD [106,107]. MMPs and their inhibitors play a role in the pathogenesis of AMD [108] [109,110] by reorganizing the extracellular matrix [111] and reduction in the build up of sub-RPE material [58]. CDKN1A is involved in RPE differentiation, proliferation, migration, and cell-cycle progression [112]. As such, CDKN1A plays a role in the ageing process of RPE and hence in AMD [113]. SERPINA1 is serine protease inhibitor and has a role in prevention against tissue destruction [114]. This protein was reported as a component of drusen [115] and its level in vitreous is directly associated with neovascular AMD and linked to increased immune response [116]. miR-145 is a biomarker for early detection of congenital hemochromatosis in AMD [117] and neovascular AMD [118]. miR-145 downregulates junctional cell adhesion molecules and actin-bundling proteins, resulting in a reduction in cell motility [119]. Tissue factor (F3) has been significantly upregulated in human macular lesions in AMD, initiating intracellular signaling and promoting inflammation and angiogenesis [120,121]. Finally, FAS-mediated apoptosis has been shown to be involved in AMD [122] via mediating RPE survival under oxidative stress conditions [123]. Therefore, it is possible that zinc might replace/substitute TGFB in eliciting a positive response in AMD.

## 5. Conclusions

In summary, we found that zinc supplementation has the potential to directly influence a number of molecular pathways. In a multi-factorial disease such as AMD, these wide-ranging effects are probably not surprising. The finding that zinc could act as a potential master regulator similar to that of TGFB1 might help to explain the benefit observed in clinical studies. Studies with a larger sample size and cells directly derived from AMD patients might help to refine our observations.

## Figures and Tables

**Figure 1 nutrients-12-03051-f001:**
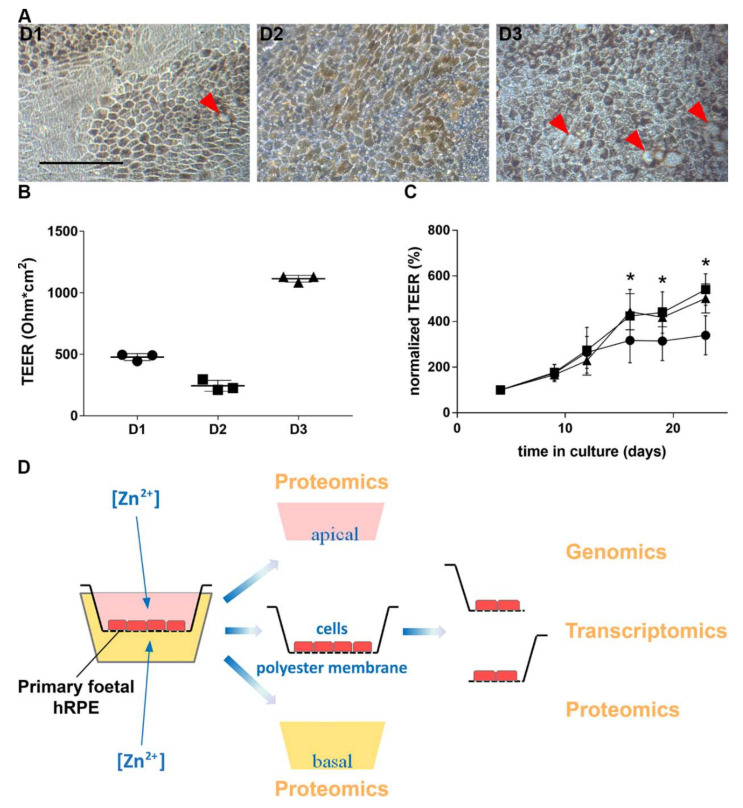
Differences in characteristics between cultures from three donors. There were differences between the three batches of cells in their pigmentation (**A**) and maximal transepithelial electrical resistance (TEER) (**B**); each point is an average +/-SEM from triplicate measurements from each of three donors.). Apical and basal zinc supplementation for 4 weeks significantly increased TEER values (**C**); the samples from the three donors represent replicates, plotted as values normalised to 100% at day 4; control (●), zinc apical supplementation (■), and zinc basal supplementation (▲). Scale bar represents 100 µm. Experimental workflow for downstream analysis (**D**). * represents *p*-value < 0.05, two-way ANOVA.

**Figure 2 nutrients-12-03051-f002:**
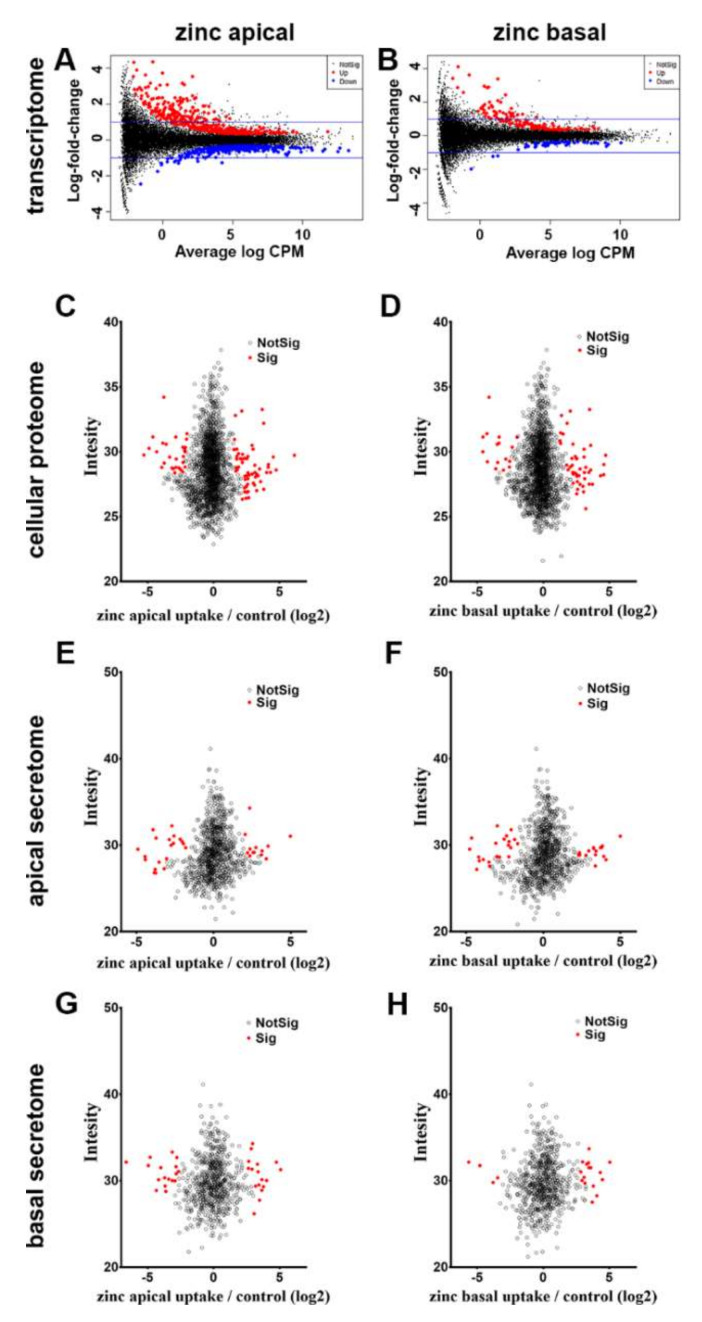
Changes in transcriptome and proteome following zinc supplementation. Comparison of cellular gene expressional after apical (**A**), or basal (**B**) zinc supplementation compared to untreated controls. Red dots represent significantly upregulated while blue dot represents significantly downregulated genes (adjusted *p*-value <0.05). Blue line indicates log2 fold change: 1 and -1. Analysis of the cellular proteome following apical (**C**) or basal (**D**) zinc supplementation. When changes in secreted proteins were analysed we found significant changes in both the apical (**E**,**F**) and basal (**G**,**H**) secretomes following apical (**E**,**G**) or basal (**F**,**H**) supplementation with zinc. In C–H, open black circles represent no change while red dots show significant significance B changes (adjusted *p*-value < 0.05).

**Figure 3 nutrients-12-03051-f003:**
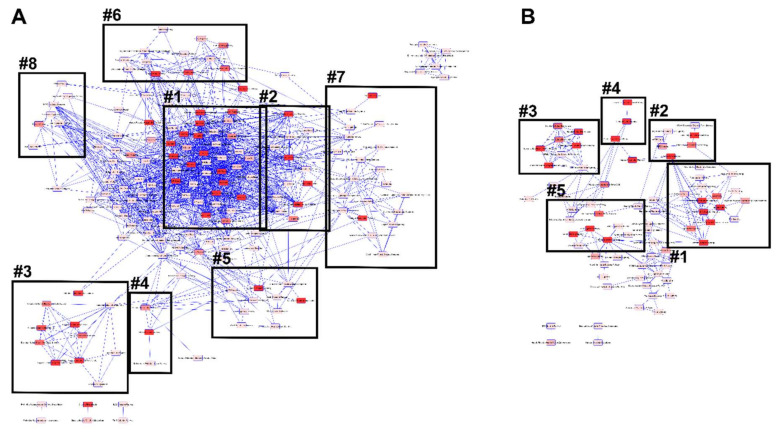
Overlapping canonical pathways upon zinc apical (**A**) or basal (**B**) supplementation. Each pathway is a single “node” colored according to significance (the brighter red, the more significant the pathway). Blue lines connect any two pathways which have at least three dataset molecules in common. GROUP #1: adhesion to polarity; GROUP #2: protein export/import; GROUP #3: extracellular matrix organization, GROUP #4: protein processing, GROUP #5: oxidative stress response, GROUP #6: embryonic cell development, GROUP #7: immune response, and GROUP #8: lipid metabolism.

**Figure 4 nutrients-12-03051-f004:**
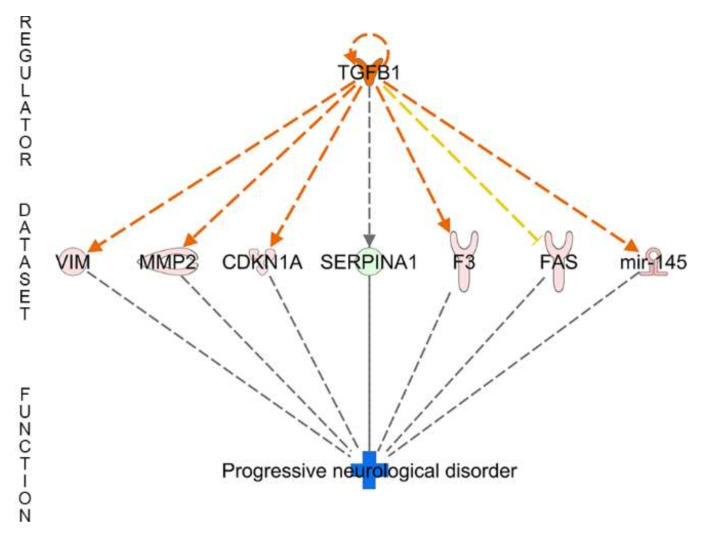
Influence of zinc supplementation culturing condition in retinal pigment epithelial cells. Measured and predicted upstream and downstream regulatory effect. Orange lines and objects: predicted activation; grey lines: not predicted effect; yellow lines: inconsistency with state of downstream molecule; light red objects: measured increase; green object: measured decrease; blue object: predicted inhibition. Note, this regulatory effect of TGFB1 was only detected upon zinc apical supplementation.

**Table 1 nutrients-12-03051-t001:** The top five hits of DAVID analysis in response to zinc supplementation.

Zinc Apical Supplementation				Zinc Basal Supplementation			
Category/Term	Count	*p*-Value	Benjamini–Hochberg	Term	Count	*p*-Value	Benjamini–Hochberg
**Biological process**				**Biological process**			
GO:0030049~muscle filament sliding	12	0	0.003	GO:0030198~extracellular matrix organization	14	0	0.0735
GO:0051603~proteolysis involved in cellular protein catabolic process	13	0	0.0027	GO:0030334~regulation of cell migration	9	0	0.0386
GO:0001501~skeletal system development	21	0	0.0075	GO:0033627~cell adhesion mediated by integrin	5	0.0001	0.0617
GO:0030198~extracellular matrix organization	25	0	0.0153	GO:2000427~positive regulation of apoptotic cell clearance	4	0.0002	0.0752
GO:0090090~negative regulation of canonical Wnt signaling pathway	22	0	0.0187	GO:0051603~proteolysis involved in cellular protein catabolic process	7	0.0002	0.0621
**Cellular component**				**Cellular component**			
GO:0070062~extracellular exosome	232	0	0	GO:0070062~extracellular exosome	120	0	0
GO:0031012~extracellular matrix	48	0	0	GO:0031012~extracellular matrix	29	0	0
GO:0005615~extracellular space	119	0	0	GO:0005615~extracellular space	60	0	0
GO:0005925~focal adhesion	49	0	0	GO:0005925~focal adhesion	28	0	0
GO:0005578~proteinaceous extracellular matrix	38	0	0	GO:0043209~myelin sheath	17	0	0
**Molecular function**				**Molecular function**			
GO:0003779~actin binding	33	0	0.0021	GO:0005178~integrin binding	11	0	0.0075
GO:0005509~calcium ion binding	61	0	0.0034	GO:0001948~glycoprotein binding	8	0.0001	0.0326
GO:0001948~glycoprotein binding	14	0	0.0024	GO:0098641~cadherin binding involved in cell-cell adhesion	16	0.0002	0.0306
GO:0005200~structural constituent of cytoskeleton	16	0.0002	0.04	GO:0004298~threonine-type endopeptidase activity	5	0.0004	0.051
GO:0008083~growth factor activity	20	0.0002	0.036	GO:0051015~actin filament binding	10	0.0005	0.0499
**INTERPRO domain**				**INTERPRO domain**			
IPR013098:Immunoglobulin I-set	25	0	0	IPR001781:Zinc finger, LIM type	8	0.0002	0.1089
IPR009030:Insulin-like growth factor binding protein, N-terminal	20	0	0.0125	IPR001353:Proteasome, subunit alpha/beta	5	0.0002	0.0731
IPR000867:Insulin-like growth factor-binding protein, IGFBP	8	0	0.0086	IPR003096:Smooth muscle protein/calponin	4	0.0006	0.1373
IPR024079:Metallopeptidase, catalytic domain	14	0.0001	0.0246	IPR016050:Proteasome, beta-type subunit, conserved site	4	0.0008	0.1358
IPR001353:Proteasome, subunit alpha/beta	7	0.0001	0.0401	IPR024079:Metallopeptidase, catalytic domain	7	0.002	0.2475
**KEGG_PATHWAY**				**KEGG_PATHWAY**			
hsa04978:Mineral absorption	12	0	0.0038	hsa04141:Protein processing in endoplasmic reticulum	12	0.0008	0.1423
hsa04145:Phagosome	20	0.0004	0.0526	hsa04145:Phagosome	11	0.0012	0.1044
hsa04350:TGF-beta signaling pathway	12	0.0053	0.3585	hsa04978:Mineral absorption	6	0.0022	0.1251
hsa03050:Proteasome	8	0.0086	0.4214	hsa05410:Hypertrophic cardiomyopathy (HCM)	7	0.0058	0.2364
hsa04670:Leukocyte transendothelial migration	14	0.0089	0.3623	hsa05414:Dilated cardiomyopathy	7	0.0083	0.2655

**Table 2 nutrients-12-03051-t002:** The top five hits of IPA analysis in response to zinc supplementation.

Zinc Apical Supplementation	Zinc Basal Supplementation
**Canonical Pathways**	*p*-value	Overlap	**Canonical Pathways**	*p*-value	Overlap
ILK Signaling	6.00 × 10^7^	13.5% 25/185	Huntington’s Disease Signaling	4.48 × 10^6^	6.4% 15/234
Axonal Guidance Signaling	1.80 × 10^6^	9.4% 44/467	Unfolded Protein Response	2.27 × 10^5^	12.7% 7/55
Hepatic Fibrosis/Hepatic Stellate Cell Activation	5.33 × 10^6^	12.6% 23/182	Epithelial Adherens Junction Signaling	1.28 × 10^4^	6.7% 10/150
Epithelial Adherens Junction Signaling	5.33 × 10^6^	13.3% 20/150	Antigen Presentation Pathway	2.97 × 10^4^	13.2% 5/38
Actin Cytoskeleton Signaling	2.38 × 10^5^	11.3% 24/213	Protein Ubiquitination Pathway	3.25 × 10^4^	4.9% 13/268
**Upstream Regulators**	*p*-value	Predicted activation	**Upstream Regulators**	*p*-value	Predicted Activation
IgG	1.17 × 10^8^		MRTFB	5.96 × 10^7^	
FSH	1.06 × 10^6^		TP63	1.88 × 10^6^	
MRTFB	3.47 × 10^6^	Activated	FSH	2.66 × 10^6^	
TEAD1	3.65 × 10^6^		TEAD2	2.39 × 10^5^	
TEAD4	3.65 × 10^6^		TEAD3	2.79 × 10^5^	
**Diseases and Disorders**	*p*-value range	# Molecules	**Diseases and Disorders**	*p*-value range	# Molecules
Neurological Disease	7.89 × 10^4^–4.57 × 10^18^	315	Endocrine System Disorders	6.67 × 10^4^–1.30 × 10^11^	235
Cancer	7.89 × 10^4^–4.69 × 10^15^	862	Organismal Injury and Abnormalities	7.23 × 10^4^–1.30 × 10^11^	312
Organismal Injury and Abnormalities	7.89 × 10^4^–4.69 × 10^15^	873	Hereditary Disorder	7.23 × 10^4^–1.58 × 10^10^	95
Gastrointestinal Disease	7.70 × 10^4^–9.48 × 10^14^	774	Neurological Disease	6.61 × 10^4^–1.58 × 10^10^	102
Cardiovascular Disease	6.50 × 10^4^–3.14 × 10^13^	203	Skeletal and Muscular Disorders	7.23 × 10^4^–1.58 × 10^10^	116
**Molecular and Cellular Functions**	*p*-value range	# Molecules	**Molecular and Cellular Functions**	*p*-value range	# Molecules
Cellular Movement	4.19 × 10^4^–7.88 × 10^19^	189	Cellular Movement	2.47 × 10^4^–1.06 × 10^9^	64
Molecular Transport	7.25 × 10^4^–6.55 × 10^9^	108	Protein Synthesis	1.31 × 10^6^–6.82 × 10^8^	40
Cellular Assembly and Organization	5.34 × 10^4^–3.26 × 10^8^	105	Cellular Compromise	7.45 × 10^6^–2.64 × 10^7^	29
Protein Synthesis	1.12 × 10^5^–4.12 × 10^7^	81	Molecular Transport	5.71 × 10^4^–2.64 × 10^7^	44
Cell Death and Survival	6.87 × 10^4^–9.45 × 10^7^	132	Cell Death and Survival	4.86 × 10^4^–3.59 × 10^7^	58
**Physiological System Development and Function**	*p*-value range		**Physiological System Development and Function**	*p*-value range	# Molecules
Cardiovascular System Development and Function	6.97 × 10^4^–2.19 × 10^10^	94	Cardiovascular System Development and Function	1.58 × 10^4^–1.74 × 10^7^	42
Organismal Development	6.97 × 10^4^–2.19 × 10^10^	118	Organismal Development	5.68 × 10^4^–1.74 × 10^7^	41
Tissue Development	6.97 × 10^4^–4.23 × 10^7^	132	Organ Morphology	1.20 × 10^4^–6.53 × 10^7^	9
Skeletal and Muscular System Development and Function	5.80 × 10^4^–4.99 × 10^6^	31	Tissue Development	5.68 × 10^4^–2.49 × 10^4^	26
Tissue Morphology	1.93 × 10^5^–1.93 × 10^5^	12	Connective Tissue Development and Function	5.68 × 10^4^–2.84 × 10^4^	19
**Regulator Effect Networks**	Disease and Functions	Consistency Score	**Regulator Effect Networks**	disease and functions	Consistency Score
TGFB1	Progressive neurological disorder	0	-		
**Networks**	Score		**Networks**	Score	
Cellular Movement, Immune Cell Trafficking	33		Amino Acid Metabolism, Post-Translational Modification, Small Molecule Biochemistry	41	
Cell Cycle, Cellular Development	33		Cellular Movement, Cell Death and Survival, Cell-To-Cell Signaling and Interaction	37	
Cell Death and Survival, Molecular Transport	31		Cardiovascular System Development and Function, Organ Development, Organ Morphology	32	
Cancer, Organismal Injury and Abnormalities, Reproductive System Disease	31		Cellular Movement, Cell-To-Cell Signaling and Interaction	18	
Cellular Development, Cellular Growth and Proliferation	27		Endocrine System Disorders, Gastrointestinal Disease, Metabolic Disease	16

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
