# Peer review of "A Multi-Omics Approach Identifies Key Regulatory Pathways Induced by Long-Term Zinc Supplementation in Human Primary Retinal Pigment Epithelium"

_nutrients, 2020, doi:10.3390/nu12103051_

Round 1

Reviewer 1 Report

In this study the authors use AMD model cell cultures to demonstrate protein response in vitro following exposure to Zinc. It has been shown in clinical trials that Zinc micronutrient supplementation reduces the risk of vision loss from nonneovascular age-related macular degeneration. In this piece, the authors show convincing evidence that RPE culture cells respond by artificial exposure to Zinc augments protein secretion as well as structural and adhesive proteins, anatomic changes more typical of wild type RPE, and increase permeability. All of these findings would in theory be part of the overall protective features of Zinc micronutrient supplementation. 

Author Response

Response to Reviewer 1,

We would like to thank Reviewer 1 for the positive assessment of our manuscript.

Reviewer 2 Report

In their manuscript „A Multi-OMICs Approach Identifies Key Regulatory Pathways Induced by Long-Term Zinc  Supplementation in Human Primary Retinal Pigment Epithelium“ the authors present an interesting study, using novel and state-of-the-art approaches to gain additional insight on the molecular role of zinc in different pathways counteracting age-related macular degeneration. Overall, the experiments seem well planned and performed and the manuscript is well written – only some minor adaptations are recommended:

Line 32 should be “…from 3 genetically different human donors.”

In the literature, “Trans-epithelial electrical resistance (TEER)” is more common than “trans-epithelial resistance (TER)”. The authors might want to consider changing this.

Line 79 “…media was…”

Legend to Fig 1: Please indicate the time in culture for panel b. Please also add information which statistical test has been used to identify the significances in 1C.

The usefulness of fig 3 seems rather limited, as resolution does not allow readers to identify the gene names. Maybe it would be possible to include a larger version, which is actually readable, as supplementary material.

Line 450: “TNF-a, IL-1b” should be “TNF-alpha, IL-1beta”

The supplementary tables are in a file named “non-published”. This might indicate that these files would be provided for review purpose only, but not published with the final manuscript? As these tables are referenced within the main text, I strongly suggest that these will be published alongside the manuscript.

Author Response

Response to Reviewer 2,

We would like to thank to Reviewer 2 for carefully reading our manuscript and the invaluable comments. Please see below our responses highlighted in yellow, and with track changes throughout the manuscript.

1. Line 32 should be “…from 3 genetically different human donors.”

Thank you very much for these suggestions, this change has been made.

2. In the literature, “Trans-epithelial electrical resistance (TEER)” is more common than “trans-epithelial resistance (TER)”. The authors might want to consider changing this.

Thank you very much for these suggestions, these changes have been made throughout the document.

3. Line 79 “…media was…”

Thank you very much for these suggestions, this change has been made.

4. Legend to Fig 1: Please indicate the time in culture for panel b. Please also add information which statistical test has been used to identify the significances in 1C.

Thank you very much for these suggestions, these changes have been made and a sentence was inserted in the methods section as well (line:77-79).

5. The usefulness of fig 3 seems rather limited, as resolution does not allow readers to identify the gene names. Maybe it would be possible to include a larger version, which is actually readable, as supplementary material.

We agree with the Reviewer that in the manuscript file the resolution of the image is low, and the writing is ineligible. However, the resolution of the original uploaded image is high resolution and after zooming on that image the writing becomes readable. We would like to ask the Editorial Team to help us identify how best we could present this complex figure. It is important to emphasize that the most important message from this figure is the ability to identify similar and different clusters, which is well represented even in the low-resolution WP file. In addition, the supplementary table 7 contains the list of canonical pathways that fall into the different clusters.

6. Line 450: “TNF-a, IL-1b” should be “TNF-alpha, IL-1beta”

 Thank you very much for these suggestions, this change has been made.

7. The supplementary tables are in a file named “non-published”. This might indicate that these files would be provided for review purpose only, but not published with the final manuscript? As these tables are referenced within the main text, I strongly suggest that these will be published alongside the manuscript.

We would like to discuss this with the Editorial Team as our understanding is that the supplementatry material will be available on line.